# Meta-Learning for Segmentation of *In Situ* Hybridization Gene Expression Images

**Charissa Poon**[1]                        CHARISSA.POON@RIKEN.JP
[1] *Brain Image Analysis Unit, RIKEN Center for Brain Science, Wako City, Japan*
**Michal Byra**[1,2]                           MBYRA@IPPT.PAN.PL
[2] *Institute of Fundamental Technological Research, Polish Academy of Sciences, Warsaw, Poland*
**Tomomi Shimogori**[3]                 TOMOMI.SHIMOGORI@RIKEN.JP
[3] *Laboratory for Molecular Mechanisms of Brain Development, RIKEN Center for Brain Science, Wako City, Japan*

**Henrik Skibbe**[1]                       HENRIK.SKIBBE@RIKEN.JP

**Editors:** Accepted for publication at MIDL 2024

## Abstract

Segmentation of biomedical images is often ambiguous and complicated by noise, varying contrasts, and imaging artifacts. We address the challenge of segmenting images of brain tissue in which gene expression has been localized using *in situ* hybridization. Since gene expression patterns differ widely between genes, it can be difficult to correctly discriminate pixels positive for gene expression. In testing different segmentation networks, we observed that each network had its own trade-offs between sensitivity and precision. To exploit the benefits of all trained networks, we developed a meta-network that learns to combine multiple segmentation maps from diverse segmentation architectures to generate a final segmentation that best matches the ground-truth label. In our experiments, the meta-network outperforms ensembles that simply average segmentation maps.

**Keywords:** meta-learning, segmentation, gene expression

## 1. Introduction

The brain is rich in cell diversity. Cell populations can be characterized by clustering cells based on differential gene expression. Gene expression brain atlases are invaluable resources for understanding cell diversity. *In situ* hybrization (ISH) is the gold standard technique for localizing gene expression at a single-cell resolution in fixed tissue sections. To create a gene expression brain atlas from ISH images, segmentation is necessary. However, variations in gene expression patterns as well as data acquisition parameters, such as tissue processing methods, animal age, and microscope settings, result in large variations in staining intensity and profiles between ISH images, presenting a challenging task for segmentation (Figure 1). Thus, while it is possible to segment ISH images using manual thresholding techniques or hand-engineered features, results will be subject to human bias and error. Our proposed method uses a deep learning network to combine several segmentation maps, which were themselves generated by deep learning networks, which may be useful for segmenting images in which signals are ambiguous.

Previous efforts to segment ambiguous signals from biomedical images include nnU-Net (Isensee et al., 2020), which automatically configures its own parameters, probabilistic U-Net (Kohl et al., 2018), which generates a distribution of segmentations for each input, and

network ensembles. However, nnU-Net and probabilistic U-Net require use of the specified architectures in isolation, and ensembles have limited discriminative power.

We propose a simple meta-learning network that learns to combine multiple segmentation maps into a final segmentation map (Figure 1). We show that the meta-learning network results in better segmentations than simply averaging the segmentation maps (ensembles). This network can be used as a postprocessing step following generation of segmentation maps for ambiguous images (Figure 2).

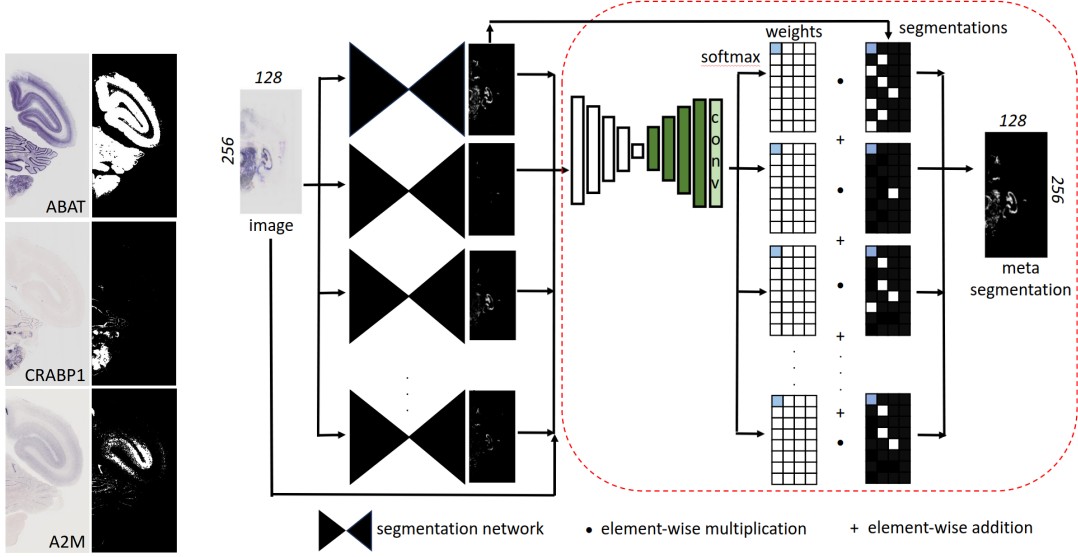

Figure 1: Left: Examples of ISH images from 3 genes and corresponding ground truth labels. Right: Schematic of Meta-Net (dotted red line).

## 2. Methods

The meta-learning network (Meta-Net) is a five-level U-Net (Ronneberger et al., 2015). Meta-Net takes as input the original ISH image and multiple corresponding segmentation maps to create pixel-level weight matrices. The weight matrices are softmaxed on a per-pixel level across all segmentation maps, and a dot product is applied between the resulting matrices and the corresponding segmentation maps to produce a final segmentation map.

We used ISH images of the neonate marmoset brain from 16 genes (947 images) in a 4:2:4 train:validation:test split, obtained from the Marmoset Gene Atlas (Shimogori et al., 2018), which is created by the Molecular Mechanisms for Brain Development lab at the RIKEN Center for Brain Science, Japan. Ground-truth labels were created using the GePS system, a grid-based manual thresholding system (Kita et al., 2021). In our experience, GePS produces adequate labels for images with consistent and minimal noise.

ISH images of single hemispheres of coronal brain sections were used, with a pixel size of 1440x840 and a pixel resolution of $\sim 18$ $\mu$m/pixel. Images, labels, and segmentations were downscaled to 256x128 pixels for processing. To correct for variations in intensity and colour

Table 1: Evaluation of segmentations of predictions from Meta-Net compared to averaging all predictions (mean ± sd, paired t-tests).

|  | Meta-Net | Ensembles | t-statistic | p-value |
|---|---|---|---|---|
| dice | 0.410 ± 0.161 | 0.388 ± 0.172 | 4.826 | 2e-06 |
| precision | 0.495 ± 0.263 | 0.484 ± 0.262 | 6.273 | 1e-09 |
| recall | 0.194 ± 0.108 | 0.161 ± 0.092 | 14.099 | 3e-36 |
| f1 | 0.260 ± 0.139 | 0.223 ± 0.125 | 14.705 | 1e-38 |

profiles between images within the same dataset, we applied histogram matching as a pre-processing step. Augmentations for colour, brightness, contrast, and hue were then stochastically applied. Each image had 25 segmentation maps generated from 25 models from 5 unique architectures: 2D and 3D U-Net (Ronneberger et al., 2015), and 2D and 3D Swin-UNETR (Hatamizadeh et al., 2022) networks, from (MONAI Consortium, 2023) and developed in-house. Inputs to 3D networks were concatenated 2D sections. We chose U-Net for its simple design but impressive performance in biomedical image segmentation tasks, and SwinUNETR to assess if its incorporation of long range information would result in better segmentations. Our code is available at https://github.com/BrainImageAnalysis/MetaNet.

## 3. Results

We evaluated the results of Meta-Net using the Dice metric, precision, recall, and the F1 score (Table 1). Segmentation maps generated from Meta-Net were superior to averaging all prediction maps across all scores quantitatively and qualitatively (Table 1, Figure 2).

## 4. Conclusion

Our results show preliminary evidence that a carefully designed meta-learning network can be used to combine segmentation maps to create a superior segmentation compared to ensembling, which is particularly important for biomedical images with ambiguous signals. Meta-Net can be easily used as a postprocessing step following generation of segmentation maps from any segmentation model. This may be useful as it is becoming increasingly simple to test diverse models using tools such as MONAI (MONAI Consortium, 2023).

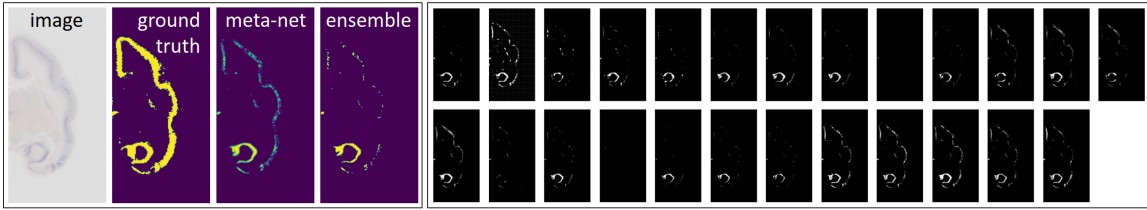

Figure 2: Left: Qualitative results. Right: Segmentation maps from single models are shown in black-and-white; ambiguity in labelling can be observed. Gene: LRRN3.

## Acknowledgments

We thank the Laboratory for Molecular Mechanisms of Brain Development at RIKEN Center for Brain Science, Japan, for curating and providing ISH data. This work was supported by the Brain Mapping by Integrated Neurotechnologies for Disease Studies (Brain/MINDS, JP15dm020700) and Multidisciplinary Frontier Brain and Neuroscience Discoveries (Brain/MINDS 2.0, JP23wm0625001) programs from the Japan Agency for Medical Research and Development AMED and JSPS Kakenhi (JP22K15658).

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
