# OpenReview forum: "Meta-Learning for Segmention of In Situ Hybridization Gene Expression Images"
_MIDL.io/2024/Short_Papers — MIDL 2024 Short Papers_

### Official Review · Reviewer_Ty6g · 2024-04-24

**Confidence:** 4
**Final Rating:** 3.5

**Review:**

The authors present a meta-learning method for ISH image segmentation. In short: given a bunch of nets trained independently, one combines their outputs with a CNN that estimates the (multiplicative) weight of each prediction at each pixel. The paper is well written, and the results are comprehensive (lots of work clearly went into this paper). The main drawbacks are: 1. limited novelty (there is plenty of literature in learning to aggregate predictions that the authors did not discuss); and 2. tiny improvement with respect to straight averaging (0.1 effect size in Table 1, which does not report p values)

Comments:

Given the relatively low Dice / poor qualitative results (Figure 2), the authors may want to emphasize how hard ISH segmentation is.

Would it make sense to train everything end to end? (both the base segmentation networks and the aggregator network)

Would it make sense to use the original resolution of the images (or at least not dowsample as much), even if this required using networks with more resolution levels?

---

### Decision · Program_Chairs · 2024-04-26

Accept